# Systemic Neoadjuvant and Adjuvant Therapies in the Management of Hepatocellular Carcinoma—A Narrative Review

**DOI:** 10.3390/cancers15133508

**Published:** 2023-07-05

**Authors:** Shadi Chamseddine, Michael LaPelusa, Ahmed Omar Kaseb

**Affiliations:** 1Department of Gastrointestinal Medical Oncology, MD Anderson Cancer Center, Houston, TX 77030, USA; schamseddine@mdanderson.org; 2Division of Cancer Medicine, MD Anderson Cancer Center, Houston, TX 77030, USA; mblapelusa@mdanderson.org

**Keywords:** hepatocellular carcinoma, neoadjuvant, adjuvant, perioperative, systemic therapy, immunotherapy

## Abstract

**Simple Summary:**

Hepatocellular carcinoma (HCC) is the most common type of liver cancer, accounting for approximately 85–90% of all cases of liver cancer worldwide. The five-year tumor recurrence rate has been estimated to be 70% among patients who present with resectable disease and undergo resection. Currently, there are no approved neoadjuvant (before surgery) or adjuvant therapies (after surgery) for these patients. This review summarizes the data from clinical trials that have evaluated systemic therapies’ safety and efficacy in the neoadjuvant and adjuvant setting for patients with resectable and potentially resectable disease.

**Abstract:**

The burden of hepatocellular carcinoma (HCC) continues to pose a significant global health problem. Several systemic therapies have recently been shown to improve survival for patients with unresectable disease. However, evidence to support the use of neoadjuvant or adjuvant systemic therapies in patients with resectable disease is limited, despite the high risk of recurrence. Neoadjuvant and adjuvant systemic therapies are being investigated for their potential to reduce recurrence after resection and improve overall survival. Our review identified various early-phase clinical trials showing impressive preliminary signals of pathologic complete response in resectable disease, and others suggesting that neoadjuvant therapies—particularly when combined with adjuvant strategies—may convert unresectable disease to resectable disease and cause significant tumor necrosis, potentially decreasing recurrence rates. The role of adjuvant therapies alone may also play a part in the management of these patients, particularly in reducing recurrence rates. Heterogeneity in trial design, therapies used, patient selection, and a scarcity of randomized phase III trials necessitate the cautious implementation of these treatment strategies. Future research is required to identify predictive biomarkers, optimize the timing and type of therapeutic combinations, and minimize treatment-related adverse effects, thereby personalizing and enhancing treatment strategies for patients with resectable and borderline resectable HCC.

## 1. Introduction

Hepatocellular carcinoma (HCC) is the most common type of liver cancer, accounting for approximately 85–90% of all cases of liver cancer worldwide [1]. The global incidence of HCC is rising, with an estimated 782,000 new cases and 745,000 deaths yearly [2]. Patients who develop HCC often have underlying chronic liver disease or cirrhosis, most frequently due to chronic viral hepatitis B or C, regular alcohol consumption, nonalcoholic fatty liver disease, and nonalcoholic steatohepatitis [3]. In the United States of America, most patients are stratified into one of four categories: localized surgical, which includes resectable or transplantable; localized non-surgical, which is liver-confined but does not meet resection or transplant criteria and is managed by local therapies alone; advanced stage with major vascular invasion on imaging and/or metastatic disease, which is amenable for systemic therapy; and finally, patients with poor performance status and or poor liver condition, defined as Child-Pugh C, who are managed best by supportive care if they are not transplant candidates.

Notably, most patients with HCC present with unresectable disease; therefore, these patients are not amenable to curative-intent treatments. However, around 10–20% of patients with HCC are diagnosed at an early stage with low tumor volume (generally a single nodule less than or equal to five centimeters in diameter or three nodules less than or equal to three centimeters in diameter that are amenable to surgical resection or transplant), with no major vascular invasion on imaging or metastasis [4]. While liver transplant affords the ability to treat both a patient’s malignancy and their chronic liver disease, a large number of patients are ineligible for transplant, and there is a scarcity of donors worldwide [5]. Therefore, surgical resection remains the most accessible option for most patients with early-stage disease that is resectable. In case a tumor does not meet resection criteria, a few loco-regional therapies may be used, with options available such as radiofrequency ablation (RFA) in early-stage HCC, and trans-arterial chemoembolization (TACE) in intermediate-stage HCC. Other options are also available, such as radioembolization with ytrrium-90 and stereotactic body radiation therapy [6]. In patients with preserved liver function diagnosed with resectable early-stage disease, resection is associated with five-year survival rates of approximately 70% [7]. However, in patients who do undergo resection, the five-year tumor recurrence rate has been estimated to be 70%, which adversely affects their survival outcome and quality of life [8]. Nevertheless, there are no approved neoadjuvant or adjuvant therapies to lower the recurrence rate. Figure 1 outlines the current HCC treatment guidelines. 

Over the past decade, several systemic therapies have improved survival for patients with unresectable disease. However, there is a scarcity of evidence to support the use of neoadjuvant or adjuvant systemic therapies in resectable disease [9]. The National Cancer Institute (NCI) defines neoadjuvant therapy as ‘treatment given as a first step to shrink a tumor before the main treatment, which is usually surgery, is given’. The NCI defines adjuvant therapy as ‘additional cancer treatment given after the primary treatment to lower the risk that the cancer will come back’. Although surgery and liver transplant remain the mainstay of therapies for patients with early-stage disease, significant challenges exist. Most patients are unsuitable for liver surgery or transplant due to tumor macrovascular invasion, multifocal disease, large tumor mass, or impaired liver function [10]. While most guidelines do not advocate for systemic therapies in the neoadjuvant or adjuvant setting based on mounting evidence of negative studies during the chemotherapy and targeted therapy era, many clinical trials integrating newer perioperative therapeutic modalities, including immunotherapies for patients with resectable and potentially resectable disease, are underway and have been recently completed. 

This review summarizes the data from recently completed clinical trials that evaluated systemic therapies’ safety and efficacy in the neoadjuvant and adjuvant setting for patients with resectable and potentially resectable disease, with a special focus on immunotherapy approaches. We also outline clinical trials in this space that are ongoing. 

## 2. Materials and Methods

To account for the changing landscape of natural history based on available hepatitis B and C therapies, as well as the emergence of metabolic syndrome and non-alcoholic steatohepatitis as major risk factors for HCC, a literature search was performed to identify phase I, II, and III trials of neoadjuvant and adjuvant therapies in patients with HCC, published between 1 January 2010 and 2023. MEDLINE (via PubMed) and ClinicalTrials.gov were searched for relevant studies, and numerous trials were identified. The search strategy used keywords including hepatocellular carcinoma, liver cancer, neoadjuvant, adjuvant, locoregional therapy, chemotherapy, systemic therapy, targeted therapy, immunotherapy, phase I, phase II, and phase III. Search results were imported into EndNote X 9.1 (Table 1).

## 3. Discussion

### 3.1. Preoperative Neoadjuvant Systemic Therapies

Limited evidence exists regarding the utility of systemic therapies in the preoperative management of resectable and potentially resectable HCC. However, some early-phase trials have evaluated these therapies in this setting (Table 2). Patients with large tumors or microvascular invasion are more likely to experience recurrence after resection [11]. Therefore, neoadjuvant therapy may be a strategy to not only downsize tumors for eventual resection, but also to minimize the chance of recurrence by eliminating sites of microscopic metastatic disease. Neoadjuvant systemic therapy, particularly via immune checkpoint inhibition, may also reduce the chance of recurrence by priming the immune system to recognize abnormal cells that exist after resection [11]. High levels of PD-L1 expression by tumor cells (and immune cells within the microenvironment) have been shown to predict recurrence after resection [12]. Introducing immune checkpoint inhibitors in the neoadjuvant setting, where a larger tumor burden exists compared to the postoperative setting, can theoretically induce a stronger T-cell response, leading to a more profound anti-tumor effect [13]. A further rationale supporting the use of immune checkpoint inhibitors in the neoadjuvant setting compared to the adjuvant setting is that resecting immune cell-rich lymph nodes—performed intraoperatively—may further lead to less postoperative anti-tumor immune cell activation and expansion by immune checkpoint inhibitors.

Several trials have evaluated the use of neoadjuvant systemic therapies in HCC that are initially unresectable. In a phase Ib trial of fifteen patients with locally advanced HCC who were initially ineligible for curative resection and received neoadjuvant cabozantinib (a tyrosine kinase inhibitor) and nivolumab (an antibody against PD-1), twelve (80%) patients eventually underwent surgery after re-staging. All were margin-negative resections. Five of the twelve patients who underwent surgery had resected tumors that showed greater than 90% necrosis [14]. Two observational studies in China evaluated a similar strategy of several combinations of tyrosine kinase and immune checkpoint inhibitors with varying success rates [15,16]. One of these observational studies showed that eight out of ten patients (80%) with initially unresectable HCC who received this combination in the neoadjuvant setting subsequently received salvage surgery. Seven out of ten patients (70%) experienced a partial response before surgery, and three out of ten (30%) patients experienced a complete response. Of the patients who underwent salvage surgery, the twelve-month recurrence-free survival rate was 75%. In the other report of sixty-three patients with initially unresectable HCC who received similar combinations of tyrosine kinase inhibitors and immune checkpoint inhibitors in the neoadjuvant setting, ten patients underwent resection. Of these ten patients, six had a partial response before surgery and six had a complete pathologic response.

Other trials have evaluated the use of neoadjuvant systemic therapies in HCC that is initially resectable. In a phase Ib trial of thirty-two patients with early-stage disease (seventeen of whom were evaluable, enrolled, and available for analysis at the time of the most recent publication), nivolumab and ipilimumab were administered prior to resection. Seventeen patients underwent resection. In nine patients with tissue available for analysis, seven (78%) achieved a pathologic response, with two (22%) achieving a complete response. After a median follow-up of six months, an objective response rate of 23% and a disease control rate of 92% were observed [17]. One review that included this analysis suggested that the discordance between pathologic and radiologic response in this trial highlights the need for further evaluation of appropriate endpoints [18]. 

In a phase II trial of twenty-four patients with early and intermediate-stage HCC who received neoadjuvant dovitinib followed by either RFA, TACE, resection, or Y-90 radioembolization, a response rate of 48% was observed, and all twenty-four patients underwent their pre-planned locoregional therapy. In this trial, seven patients ultimately proceeded to liver transplant [19].

Sorafenib has also been studied as a neoadjuvant therapy in patients with resectable HCC and was administered to patients (twenty-five of whom were evaluable for analysis of the study’s primary endpoints of anti-tumor activity and histologic changes). All patients experienced disease stability, and an objective response rate was seen in six (32%) patients. All patients underwent resection, with margin-negative resections in twenty-two (88%) patients. Tumor necrosis of 50% or greater was seen in 24% of cases [20].

**Table 2 cancers-15-03508-t002:** Preoperative Neoadjuvant Systemic Therapies.

Experimental Arm	Comparison Arm	Patient Population	Phase	Primary Outcome(s)	Safety	Registration	Reference
Cabozantinib plus nivolumab	None	15 patients with locally advanced/borderline resectable HCC	Ib	Number of adverse events Number of patients who completed preoperative treatment and proceeded to surgery	Grade 3 or higher treatment-related adverse events occurred in 2 patients	NCT03299946	[14]
Anti-PD-1 antibody (nivolumab, camrelizumab, pembrolizumab, or sintilimab) plus TKI (lenvatinib or apatinib)	None	63 patients with unresectable or advanced HCC	Case series	Not applicable	1 patient died from an immune-related adverse event	Not applicable	[15]
Anti-PD-1 antibody (pembrolizumab, toripalimab, or sintilimab) plus TKI (lenvatinib or apatinib)	None	10 patients with Child-Pugh class A and BCLC classification stage C	Case series	Not applicable	No patients experienced grade 3 or 4 treatment-related adverse events	Not applicable	[16]
Nivolumab plus ipilimumab	None	32 patients with early-stage, resectable HCC (17 enrolled and available for analysis at time of most recent publication)	Ib	Number of patients with an unplanned delay to surgery Safety and tolerability of nivolumab and ipilimumab	Grade 3 or treatment-related adverse events occurred in 1 patient	NCT03682276 EudraCT Number: 2018–000987-2	[17]
Dovitinib	None	25 patients with early and intermediate-stage, resectable HCC	II	Objective response rate Intratumoral blood flow changes	Grade 3 or 4 treatment-related adverse events occurred in 22 patients	EU-CTR 2011-002445-36	[19]
Sorafenib	None	30 patients with resectable HCC	II	Anti-tumor activity	Not reported	NCT01182272	[20]

Several trials evaluating the safety and efficacy of neoadjuvant therapies are ongoing for patients with both initially resectable and unresectable HCC (Table 3).

Reducing tumor burden in patients with advanced HCC whose disease is beyond the Milan criteria with the goal of future transplant is referred to as downstaging therapy. Factors such as waitlist time, tumor burden, and alpha-fetoprotein response to downstaging therapy have been shown to influence post-transplant outcomes [29]. Downstaging can be performed with surgery, systemic therapy, or locoregional therapies [30,31,32,33,34,35].

Several prospective studies and meta-analyses have been conducted to delineate the efficacy of downstaging therapy. One phase IIb/III trial of patients aged 18–65 years with hepatocellular carcinoma beyond the Milan criteria, absence of macrovascular invasion or extrahepatic spread, 5-year estimated post-transplantation survival of at least 50%, and good liver function (Child-Pugh A-B7) were recruited and underwent tumor downstaging with locoregional, surgical, or systemic therapies [36]. Five-year tumor event-free survival (76.8% vs. 18.3%; HR 0.20; 95% CI 0.07–0.57; *p* = 0.003) and five-year overall survival (77.5% vs. 31.2%; HR 0.32; 95% CI 0.11–0.92; *p* = 0.035) were improved in the transplant group compared to the control group. In a separate cohort study that compared post-liver transplant outcomes of patients with HCC beyond the Milan criteria who received downstaging therapy versus patients with HCC within the Milan criteria, ten-year post-transplant survival (52.1% vs. 61.5%) and recurrence rates (20.6% vs. 13.3%) were similar [37]. Further, the previously mentioned phase II trial, in which twenty-four patients with early and intermediate-stage HCC received neoadjuvant dovitinib plus locoregional therapy, led to seven patients receiving a liver transplant, although receipt of a liver transplant was not a primary or secondary endpoint. An essential consideration for using systemic therapy, rather than locoregional therapy, is the differing side effect profile. Specifically, immune-related adverse events and graft rejection after transplant must be contemplated when considering using immune checkpoint inhibitors as downstaging therapy.

Two important trials are underway to assess the efficacy of downstaging with neoadjuvant systemic therapies prior to liver transplant. One trial is evaluating the use of pembrolizumab plus lenvatinib in 192 patients with HCC prior to liver transplant, and has a primary endpoint of relapse-free survival [38]. The other trial is evaluating the use of camrelizumab plus apatinib in 120 patients with HCC prior to liver transplant, and has primary endpoints of objective remission rate and relapse-free survival [39].

### 3.2. Postoperative Adjuvant Systemic Therapies

Similar to preoperative neoadjuvant systemic therapy, the role of postoperative adjuvant systemic therapy is under investigation for patients with resectable and potentially resectable HCC. Multiple early-phase and late-phase trials have evaluated systemic adjuvant therapies in this setting (Table 4).

The STORM trial, which was the first large randomized controlled trial assessing adjuvant therapy in HCC, and the only one completed in the United States, assessed postoperative adjuvant therapy for patients with resected HCC involved sorafenib, a kinase inhibitor approved for treating patients with advanced HCC. Over 1000 patients underwent local ablation or surgical resection, followed by adjuvant therapy with either sorafenib or placebo. No significant improvement in relapse-free survival (RFS) was seen in the sorafenib group versus the placebo group (33.3 vs. 33.7 months; hazard ratio (HR 0.940; 95% CI 0.780–1.134; *p* = 0.26) or median overall survival (OS) (HR 0.995; 95% CI 0.761–1.300; *p* = 0.48) at a median follow-up of 23.0 months in the sorafenib group and 22.0 months in the placebo group were observed [40].

Two other large, randomized trials investigated the efficacy of adjuvant interferon alfa-2b (IFNα-2b) therapy in patients with resected HCC driven by hepatitis C, and concluded that adjuvant IFNα-2b did not reduce recurrence with a higher prevalence of treatment-related toxicity. When comparing patients treated with IFNα-2b to no adjuvant therapy, the first of these two trials resulted in no difference in RFS at a median follow-up of 45 months (24.3% vs. 5.8%; *p* = 0.49) [41]. The second of these two trials similarly showed no improvement in RFS among patients who received adjuvant IFNα-2b versus placebo (42.2 vs. 48.6 months; *p* = 0.828) [42].

Adjuvant cellular therapy in patients with HCC gained interest as the benefit conferred by immune checkpoint inhibitors to patients with HCC became apparent. A phase III trial of 230 patients with HCC treated with curative intent by resection, RFA, or percutaneous ethanol injection who received adjuvant autologous cytokine-induced killer cells or no adjuvant therapy was conducted in Korea. The median RFS was 14.0 months longer in patients who received cellular therapy than the control group (44.0 vs. 30.0 months; HR 0.63; 95% CI 0.43–0.94; *p* = 0.010) [43].

Multiple trials evaluating the safety and efficacy of postoperative adjuvant systemic therapies are ongoing for patients with resected HCC (Table 5). Notably, the IMbrave050 trial is a phase III multicenter randomized study aiming to assess atezolizumab (an antibody against PD-L1) plus bevacizumab (an antibody against vascular endothelial growth factor) versus active surveillance in patients with HCC at a high risk of recurrence following curative resection or ablation. This study included patients with segmental portal vein invasion (V1) and right anterior or posterior portal vein (V2). RFS is the primary outcome being assessed, and OS and time-to-recurrence are the secondary endpoints. It was conducted at 170 sites in twenty-five countries with the goal of recruiting 662 patients [44], and was recently reported to have met its primary endpoint of recurrence-free survival (HR = 0.72; 95% CI 0.56–0.93; *p* = 0.0120) [45].

### 3.3. Perioperative (Combined Preoperative Neoadjuvant and Postoperative Adjuvant) Systemic Therapies

In addition to the trials that have evaluated neoadjuvant or adjuvant therapies alone, there have also been trials conducted to assess the safety and efficacy of strategies utilizing both neoadjuvant and adjuvant therapies (Table 6). 

One phase II trial conducted by our group studied twenty-seven patients with resectable HCC who received neoadjuvant and adjuvant nivolumab with and without ipilimumab, an antibody against cytotoxic-lymphocyte associated protein 4 (CTLA-4). In this single-center, randomized, open-label, phase 2 trial, patients with resectable hepatocellular carcinoma were randomly divided into two groups. In the first group, patients received 240 mg of nivolumab through intravenous administration every 2 weeks. They received up to three doses before the surgery scheduled at 6 weeks. After surgery, in the adjuvant phase, they were given 480 mg of nivolumab intravenously every 4 weeks for a duration of 2 years. In the second group, patients received 240 mg of nivolumab intravenously every 2 weeks, up to three doses before the surgery. Additionally, they were administered one dose of 1 mg/kg of ipilimumab intravenously concurrently with the first preoperative dose of nivolumab. Following surgery, in the adjuvant phase, they received 480 mg of nivolumab intravenously every 4 weeks for up to 2 years, along with 1 mg/kg of ipilimumab intravenously every 6 weeks for up to four cycles. Twenty patients underwent resection, and a median PFS of 9.4 months in the nivolumab group versus 19.5 months in the nivolumab plus ipilimumab group was observed (HR 0.9; 95% CI 0.31–2.54). On histologic examination from tissue obtained at resection, six patients achieved major pathologic response, defined as >70% necrosis: three of nine patients who received nivolumab alone compared to three of eleven patients who received nivolumab plus ipilimumab had greater than 70% tumor necrosis. In fact, five out of the six patients achieved complete pathologic response (100% necrosis with no viable HCC). Remarkably, at two-year follow-up, no patients with greater than 70% tumor necrosis experienced recurrence, while approximately half of patients with less than 70% tumor necrosis experienced recurrence [54]. Notably, our group studies imaging correlatives with response to immunotherapy and reported on the correlation between tumor stiffness, assessed by elastography, and better response to immunotherapy in HCC [55].

Further research is warranted in this area to validate these results and assess if it correlates with tumor-infiltrating lymphocytes and/or increased traffic at the interface between the tumors and surrounding liver tissue. The combination of neoadjuvant and adjuvant camrelizumab, an antibody against PD-1, plus apatinib, a tyrosine kinase inhibitor, was studied in a phase II trial of eighteen patients with resectable HCC. One patient did not undergo resection. Of those who did, three patients had tumors with greater than 90% necrosis, and one had a complete pathologic response. Five patients did not receive adjuvant therapy for external reasons, and among thirteen patients who did receive adjuvant therapy, an RFS rate of 53.9% was observed. Further, fewer patients with greater than 90% tumor necrosis experienced recurrence [56]. Another antibody against PD-1, cemepilimab, was studied in the neoadjuvant and adjuvant setting in a phase II trial of twenty-one patients with resectable HCC (Table 6). Of twenty patients who underwent resection, four had greater than 70% tumor necrosis, and seven had greater than 50% tumor necrosis. This study also showed that more immune infiltration was present in pre-treatment tumor specimens among patients with a higher percentage of tumor necrosis, highlighting the potential for this histologic finding to serve as a biomarker predictive of response to HCC with immune checkpoint inhibitors [57]. Figure 2 provides a summary highlighting the future multimodal treatment strategy for resectable HCC.

**Table 6 cancers-15-03508-t006:** Combined Preoperative Neoadjuvant and Postoperative Adjuvant Systemic Therapies.

Experimental Arm	Comparison Arm	Patient Population	Phase	Primary Outcome(s)	Safety	Registration	Reference
Neoadjuvant and adjuvant nivolumab	Neoadjuvant nivolumab plus ipilimumab and adjuvant nivolumab plus ipilimumab	30 patients with resectable HCC (27 enrolled)	II	Safety and tolerability of nivolumab with or without ipilimumab	Grade 3 or higher treatment-related adverse effects occurred in 3 (23%) of patients in the nivolumab arm and 6 (43%) patients in the nivolumab plus ipilimumab arm	NCT03222076	[54]
Neoadjuvant and adjuvant camrelizumab plus apatinib	None	18 patients with resectable HCC	II	Major pathologic response (90% or greater tumor necrosis)	Grade 3 or higher treatment-related adverse effects occurred in 3 (16.7%) patients	NCT04297202	[56]
Cemepilimab	None	21 patients with resectable HCC	II	Significant tumor necrosis (>70% or greater tumor necrosis)	Grade 3 or higher treatment-related adverse effects occurred in 7 (33.3%) patients	NCT03916627	[57]

Several trials assessing neoadjuvant and adjuvant therapy in resectable HCC are ongoing (Table 7). 

## 4. Conclusions and Future Directions

The landscape of systemic therapies, particularly in the neoadjuvant and adjuvant settings, is evolving and being refined to decrease recurrence rates and improve survival [69]. This review summarizes the safety and efficacy of these investigational treatment strategies which have recently incorporated immunotherapy approaches, given their expanded role in HCC in front- and second-line setting in the advanced disease stage. Currently, the National Comprehensive Cancer Network does not provide a clear recommendation supporting the routine use of perioperative systemic therapies for patients with resectable or potentially resectable HCC [70]. 

In the previously mentioned trials, many neoadjuvant systemic therapies, both alone and when combined with adjuvant therapies, have shown the potential to downsize HCC tumors and turn unresectable disease into resectable. Furthermore, neoadjuvant therapies have been shown to cause major tumor necrosis, which may be associated with decreased recurrence rates. Less is known about the impact of adjuvant systemic therapies alone. However, according to preliminary results from the phase III IMbrave050 study, adjuvant therapy with atezolizumab and bevacizumab improved recurrence-free survival in patients with HCC following surgical resection or ablation [44]. 

Finally, while progress has been made in understanding the role of systemic therapies for patients with borderline resectable and resectable HCC, significant challenges remain. The association between tumor necrosis induced by neoadjuvant therapies and objective response and how these endpoints influence recurrence rates warrant further investigation. Notably, in the era of immunotherapy advances in HCC, neoadjuvant approach is advantageous, given the existing active tumor microenvironment with tumor dense tumor-infiltrating lymphocytes before surgery. Additionally, future research should emphasize identifying prognostic biomarkers (tissue, blood, and imaging) indicative of the initial response to neoadjuvant therapy, selecting the most appropriate combination of systemic and locoregional therapies, while maintaining a manageable safety profile, and optimizing the timing of adjuvant therapies based on radiographic, tissue-based, and serologic markers that could predict recurrence. 

## Figures and Tables

**Figure 1 cancers-15-03508-f001:**
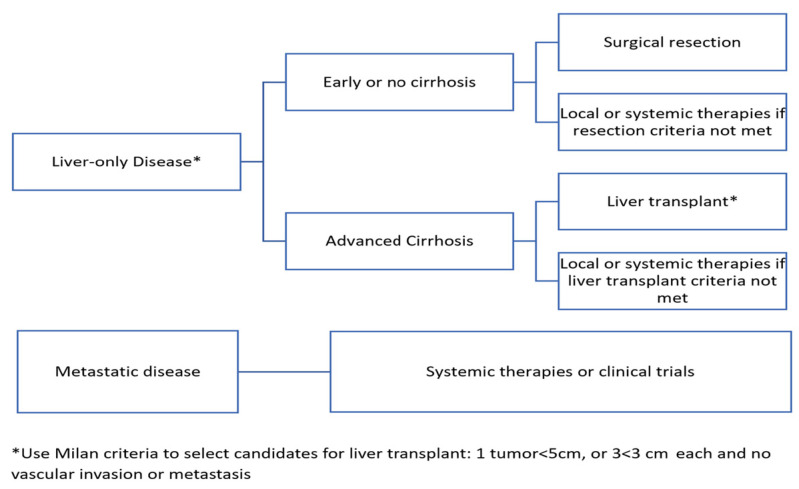
HCC Treatment Guidelines.

**Figure 2 cancers-15-03508-f002:**
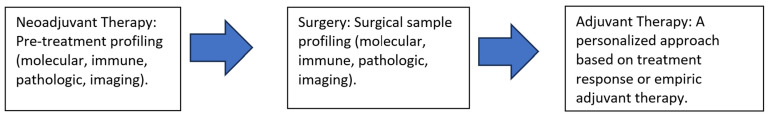
The future strategy for resectable HCC.

**Table 1 cancers-15-03508-t001:** Search Strategy.

Items	Specification
Date of search	5 January 2023
Databases and other sources searched	PubMed, ClinicalTrials.gov
Search terms used	Hepatocellular carcinoma, liver cancer, locoregional therapy, chemotherapy, targeted therapy, immunotherapy, randomized trials, controlled trials, phase I, phase II, phase III
Inclusion and exclusion criteria	Trials were excluded if they were not completed, closed early, or did not report their outcome(s) in the form of a published abstract or manuscript
Selection process	The authors conducted an independent search

**Table 3 cancers-15-03508-t003:** Ongoing Trials for Preoperative Neoadjuvant Systemic Therapies.

Experimental Arm	Comparison Arm	Patient Population	Phase	Primary Outcome(s)	Registration	Reference
Nivolumab plus ipilimumab	None	40 patients with potential for curative surgical resection	II	Percentage of patients with tumor shrinkage > 10%	NCT03510871	[21]
Camrelizumab plus apatinib mesylate plus oxaliplatin	None	15 participants with locally advanced, potentially resectable disease	II	Major pathological response (>90% tumor necrosis)	NCT04850040	[22]
Sorafenib plus capecitabine plus oxaliplatin	None	15 participants with HCC confined to a single lobe and not suitable for surgery or locoregional therapies	II	Proportion of patients with resectable disease	NCT03578874	[23]
Sintilimab plus transarterial chemoembolization	None	61 patients with BCLC stage A (not transplantable) or stage B (ineligible for resection)	II	Duration from treatment initiation to disease progression in patients who cannot undergo surgery, or to the date of relapse after surgery, or death	NCT04174781	[24]
Tislelizumab plus intensity modulated radiation therapy	None	30 patients with resectable disease and portal vein tumor thrombus	II	Relapse-free survival	NCT04850157	[24]
Sorafenib plus laser ablation	Laser ablation	40 patients with unresectable HCC containing one nodule larger than 4 cm in diameter	II	Complete tumor ablation rate, time-to-recurrence (in complete response group), time-to-progression (in partial response group)	NCT01507064	[25]
Atezolizumab plus bevacizumab plus stereotactic beam radiation therapy	None	20 patients with resectable disease	I	Proportion of patients with grade 3 or 4 treatment-related adverse events	NCT04857684	[26]
Anlotinib hydrochloride plus TQB2450 (antibody against PD-L1)	None	20 patients with resectable disease	Ib	Pathologic complete response rate, overall response rate	NCT04888546	[27]
Atezolizumab plus bevacizumab	None	30 participants with resectable HCC	II	Pathologic complete response rate Safety/tolerability	NCT04721132	[28]

Downstaging Neoadjuvant Therapies Prior to Transplant.

**Table 4 cancers-15-03508-t004:** Postoperative Adjuvant Systemic Therapies.

Experimental Arm	Comparison Arm	Patient Population	Phase	Primary Outcome	Safety	Registration	Reference
Sorafenib	No adjuvant therapy	1114 patients with HCC who received curative-intent therapy with either resection or local ablation	III	RFS	Grade 3 or 4 drug-related adverse events occurred in 293 patients in sorafenib group vs. 51 in placebo group	NCT00692770	[40]
IFNα-2b	No adjuvant therapy	150 patients with HCV-driven HCC who underwent resection	III	RFS	9 (12%) patients in IFNα-2b group experienced toxicity resulting in dose reduction, 6 of whom stopped therapy	NCT00273247	[41]
IFNα-2b	No adjuvant therapy	268 patients with HCV-driven HCC who underwent resection	III	RFS	Grade 3 or 4 adverse events related to fatigue (*p* = 0.035), leukopenia (*p* = 0.003), granulocytopenia (*p* < 0.001), and thrombocytopenia (*p* = 0.010) occurred in significantly more patients in IFNα-2b group	NCT00149565	[42]
Autologous cytokine-induced killer cells	No adjuvant therapy	230 patients with HCC who received curative-intent therapy with either resection, RFA, or percutaneous ethanol injection	III	RFS	AEs occurred more frequently in the cellular therapy group (62% vs. 41%; *p* = 0.002) The rate of grade 3 or 4 AEs was comparable between groups (7.8% vs. 3.5%; *p* = 0.15)	NCT00699816	[43]

**Table 5 cancers-15-03508-t005:** Ongoing Trials for Postoperative Adjuvant Systemic Therapies.

Experimental Arm	Comparison Arm	Patient Population	Phase	Primary Outcome	Registration	Reference
Atezolizumab plus bevacizumab	No adjuvant therapy	668 patients with HCC who have undergone curative resection or ablation	III	RFS	NCT04102098	[44]
Durvalumab plus bevacizumab	Durvalumab	908 patients with HCC who have undergone curative therapy with resection or ablation	III	RFS	NCT03847428	[46]
	No adjuvant therapy					
Nivolumab	No adjuvant therapy	545 patients with HCC who have undergone curative resection or ablation	III	RFS	NCT03383458	[47]
Pembrolizumab	No adjuvant therapy	950 patients with HCC who have undergone curative resection or ablation	III	RFS, OS	NCT03867084	[48]
Camrelizumab plus rivoceranib (apatinib)	No adjuvant therapy	687 patients with HCC who have undergone curative resection or ablation	III	RFS	NCT04639180	[49]
Camrelizumab plus rivoceranib (apatinib)	Camrelizumab	250 patients with HCC who have undergone curative resection or ablation	II	RFS	NCT05367687	[50]
Donafenib plus tislelizumab	No adjuvant therapy	32 patients with HCC who have undergone curative resection	II	1-year RFS	NCT05545124	[51]
Tislelizumab plus sitravatinib	No adjuvant therapy	40 patients with HCC who have undergone curative resection	II	2-years RFS	NCT05407519	[52]
Donafenib and anti-PD-1 antibody (unspecified)	No adjuvant therapy	30 patients with HCC who have undergone curative resection	I	1-year RFS	NCT04418401	[53]

**Table 7 cancers-15-03508-t007:** Ongoing Trials for Combined Preoperative Neoadjuvant and Postoperative Adjuvant Systemic Therapies.

Experimental Arm(s)	Comparison Arm	Patient Population	Phase	Primary Outcome	Registration	Reference
Neoadjuvant tremelimumab plus durvalumab and adjuvant durvalumab	None	28 patients with resectable HCC	II	Number of greater grade 3 or higher adverse events or immune-related adverse events that lead to treatment cessation	NCT05440864	[58]
Neoadjuvant nivolumab followed by electroporation and adjuvant nivolumab	None	43 patients with resectable HCC	II	Local recurrence-free survival during 1-year follow-up	NCT03630640	[59]
Neoadjuvant atezolizumab followed by RFA and adjuvant atezolizumab plus bevacizumab	RFA alone	202 patients with HCC eligible for ablation	II	Recurrence-free survival	NCT04727307	[60]
Neoadjuvant camrelizumab plus apatinib and adjuvant camrelizumab	Adjuvant camrelizumab	78 patients with resectable HCC	II	1-year tumor recurrence-free rate	NCT04930315	[61]
Neoadjuvant tislelizumab plus lenvatinib and adjuvant tislelizumab plus lenvatinib	N/A	30 patients with resectable HCC	II	Safety as measured by the number of grade 3 and grade 4 adverse events that occurred when subjects participated in the study, feasibility as measured by rate of enrollment	NCT04834986	[62]
Neoadjuvant nivolumab and adjuvant nivolumab	Neoadjuvant nivolumab plus relatlimab and adjuvant nivolumab plus relatlimab	20 patients with resectable HCC	I	Number of patients who complete neoadjuvant therapy and proceed to surgery	NCT04658147	[63]
Neoadjuvant pembrolizumab plus lenvatinib and adjuvant pembrolizumab	Neoadjuvant pembrolizumab or lenvatinib and adjuvant pembrolizumab	60 patients with resectable HCC	II	Major pathological response rate, defined as the proportion of patients with less than 10% viable tumor	NCT05185739	[64]
Neoadjuvant toripalimab and adjuvant toripalimab	Neoadjuvant toripalimab plus lenvatinib and adjuvant toripalimab plus lenvatinib	40 patients with resectable HCC	Ib/II	Pathological response rate	NCT03867370	[65]
	Neoadjuvant toripalimab plus lenvatinib and adjuvant toripalimab					
Neoadjuvant tislelizumab and adjuvant tislelizumab	Neoadjuvant tislelizumab plus lenvatinib and adjuvant tislelizumab plus lenvatinib	80 patients with resectable HCC	II	Disease-free survival	NCT04615143	[66]
Neoadjuvant atezolizumab plus bevacizumab and adjuvant atezolizumab plus bevacizuamb	None	45 patients with potentially resectable HCC	II	Pathologic complete response rate, distinct immunophenotypes, and dynamic changes of tumor-infiltrating cells	NCT04954339	[67]
Neoadjuvant camrelizumab followed by TACE and adjuvant camrelizumab, plus apatinib	TACE	290 patients with resectable HCC	None	Three-year event free-survival, major pathologic response rate (less than 50% residual tumor)	NCT04521153	[68]

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
