# Peer review of "Systemic Neoadjuvant and Adjuvant Therapies in the Management of Hepatocellular Carcinoma—A Narrative Review"

_cancers, 2023, doi:10.3390/cancers15133508_

Round 1
Reviewer 1 Report
The work by LaPestula et al describes a meta-analysis of the use of adjuvant and new adjuvant therapy in the management of HCC. The manuscript is well-written and provides a survey of the reported literature in the field.
The manuscript although discussing adjuvant and neoadjuvant therapy, the `authors did not define the terms used or give a brief description of the treatments and the probable mode of action of the used therapies which needs to be addressed either in the introduction section or the discussion section.
The English language is very good. I have noted two instants that need to be corrected:
1. In the abstract, the authors state "The role of adjuvant therapies alone may also have a role in 19 the management "The word role is repeated two times which needs to be corrected to make a logical flow of the statement.
2. In the conclusion section line 288, the authors state "Notably, in the era of immunotherapy advanced in HCC ", I think the word advanced needs to be changed to advances.
Reviewer 2 Report
In this review the authors summarize the data from clinical trials that study the use of systemic therapies in the neoadjuvant and adjuvant setting for HCC patients. The try to emphasize the importance of neoadjuvant to modified the paradigm of HCC recurrence.
The manuscript is well written, easy to understand and complete. However, there are some minor issues that need to be addressed:
1) Please, the first time you use an abbreviation, it’s important to spell out the full term and put the abbreviation in parentheses. In contrast, if abbreviations are not going to be use more than once time it not to be added. Please change them through the text (for instance: NASH; TTS line 218)
2) It would be nice to include a paragraph including what is the meaning of adjuvant and neoadjuvant therapies.
3) Please pay attention in the references. The are a lack of references in the paragraph line 104 to 122.
4) Although there is too many information in the tables, it would be nice to include the results get with the therapies, for instance: improve the overall survival or improve RFS: YES/NO
5) There are too many possibilities to use systemic treatments, before surgery, after surgery, both of them. It would be great to include a figure to summarize it.
6) A brief paragraph including the loco-regional treatments and systemic therapies for HCC according to clinical practice guidelines would improve the paper. It would explain the need to create new scenarios of systemic therapy for HCC (neoadjuvant/adjuvant).
Reviewer 3 Report
1. page 4/15 line 133 & 6/15 line 167, reference 18: a short 4 week medication and 5day/on , 2day/off, 83% reduced dose, 3 BCLC A and 4 BCLC B got transplant, could you describe this paper more detaily
2. about the nivolumab +/- ipilimumab , could you describe ipilimumab dose
page 4/15 reference 16
page 5/15, reference 20
page 10/15, reference 53
3. page 7/15, you could tell us this is the STORM trial
4. page 8/15 IMbrave050, VP1+2 were enrolled
4. Page 12/15 line 303, what it " *** "
5. could you provide/summarize the difference between MKI and IO in the neoadjuvant and adjuvant treatment (primary resistance or iRAE leads to tumor progression or treatment interruption) and the quality of life also is different, to guide the reader on how to explain to their patients
6. small suggestion, page 9/15, table , reference 50, 51: trisleli-zumab and si-travatinib could be present in the next line and avoiding the "-"
thanks for your review
